# Liver Progenitor Cells: Cellular Origins, Plasticity, and Signaling Pathways in Liver Regeneration

**DOI:** 10.3390/biology14101361

**Published:** 2025-10-04

**Authors:** Jinsol Han, Ahyeon Sung, Hayeong Jeong, Youngmi Jung

**Affiliations:** 1Institute of Systems Biology, Pusan National University, Pusan 46241, Republic of Korea; wlsthf1408@pusan.ac.kr; 2Department of Integrated Biological Science, College of Natural Science, Pusan National University, Pusan 46241, Republic of Korea; suarha@pusan.ac.kr (A.S.); jeong17@pusan.ac.kr (H.J.); 3Department of Biological Sciences, College of Natural Science, Pusan National University, Pusan 46241, Republic of Korea; 4Research Institute for Convergence of Biomedical Science and Technology, Pusan National University Yangsan Hospital, Yangsan 50612, Republic of Korea

**Keywords:** liver progenitor cells, liver regeneration, cellular plasticity, dedifferentiation, hepatocytes, cholangiocytes, hepatic stellate cells

## Abstract

**Simple Summary:**

The liver is one of the few organs in the body that can regrow after injury or surgery. Normally, this happens when liver cells, called hepatocytes, multiply to repair the damage. However, if hepatocytes are too damaged to regenerate, other types of liver cells start multiplying instead. One important group of these cells is called liver progenitor cells (LPCs). LPCs may originate from both outside and inside the liver. Inside the liver, cells such as hepatocytes, bile duct cells, and hepatic stellate cells can lose their specialized features and return to a more flexible state as LPCs. These LPCs can then develop into mature liver cell types again. This transformation is carefully controlled by different signaling pathways. In this review, we provide an overview of what is currently known about where LPCs come from—particularly those that originate within the liver—and the signals that control their activation and development during liver regeneration. A better understanding of how LPCs work could lead to new ways to support liver repair when normal regeneration fails, offering hope for treating serious liver injuries or diseases.

**Abstract:**

The liver has a notable regenerative capacity, primarily through hepatocyte proliferation. However, when this process is impaired—due to severe and/or chronic injury—liver progenitor cells (LPCs) serve as a facultative reserve to restore hepatic function. LPCs, which are a bipotent and heterogeneous population located near the canals of Hering, can differentiate into hepatocytes and cholangiocytes. Recent evidence suggests that LPCs may originate from mature hepatic cells—such as hepatocytes, cholangiocytes, and hepatic stellate cells—through dedifferentiation under specific injury conditions. Cellular plasticity in the liver is governed by complex signaling networks that regulate LPC activation, maintenance, and lineage commitment. However, the precise cellular origin of LPCs and the mechanisms driving their activation remain incompletely defined. Therefore, this review aims to synthesize current insights into LPC biology and emphasize their diverse cellular origins, functional roles in liver regeneration, and the key signaling pathways involved. A deeper understanding of LPC dynamics may ultimately guide the development of novel therapeutic strategies to enhance liver regeneration in chronic liver disease.

## 1. Introduction

The liver is a unique organ with exceptional regenerative ability [1,2,3]. Following substantial hepatic volume loss from partial hepatectomy (PHx)—which is the surgical removal of a portion of the liver—it rapidly restores its mass and volume [2,4]. In humans, nearly 90% of liver tissue can be reestablished within 6 months [5]. In rodents, the liver fully compensates within 7–10 days after a 70% PHx [1,6]. The regenerative process of the liver begins primarily with hepatocyte proliferation [7]. However, when hepatocyte proliferation is impaired—due to chronic injury, senescence, or genetic ablation—liver regeneration relies on alternative cellular sources [8,9]. Among these, liver progenitor cells (LPCs), a small population of undifferentiated cells in the biliary compartment of the healthy liver, have emerged as key mediators [10]. LPCs are capable of differentiating into hepatic cells, such as hepatocytes, cholangiocytes, and hepatic stellate cells (HSCs), thus contributing to tissue repair and functional recovery [2,11,12].

Despite increasing recognition of LPCs in liver biology and disease, their precise origins and the mechanisms underlying their activation and regulation remain unclear. Recent advances, including genetic lineage tracing and single-cell transcriptomics, reveal new insights into the cellular plasticity of hepatocytes, cholangiocytes, and HSCs, suggesting that LPCs may emerge from the transition of mature hepatic cells under specific injury-induced conditions [13,14,15,16,17,18]. These transitions are controlled by complex signaling pathways and microenvironmental cues, which also guide LPC redifferentiation into functional hepatic cells during tissue repair [10,19]. This review summarizes current understanding of the cellular origins of LPCs and the molecular mechanisms governing their bidirectional transitions. We focus on the signaling pathways that drive intrahepatic cell conversion into LPCs, as well as those regulating LPC differentiation into functional hepatic cells. By integrating experimental and clinical findings, we aim to provide a comprehensive perspective on LPC plasticity and its regulation, highlighting the dynamic interplay between cellular identity and liver regeneration.

## 2. General Information on Liver Progenitor Cells

LPCs are small cells with a prominent oval nucleus and scant cytoplasm [20,21]. They have been described under various names, including oval cells, hepatic progenitor cells, liver stem cells, and atypical ductular cells [21,22]. Despite this nomenclatural diversity, these terms all refer to a bipotent epithelial cell population [23,24]. LPCs are primarily concentrated in the portal triad, particularly areas adjacent to the canals of Hering [25,26]. These canals form the peripheral branches of the biliary tree and link hepatocyte canaliculi to interlobular bile ducts [27,28]. As the interface between hepatocytes and cholangiocytes, the canals of Hering constitute a pivotal niche for LPCs, which can differentiate into both epithelial lineages [27,28].

As with other stem cell populations, LPCs are heterogeneous and cannot be identified by a single marker [29]. Given their differentiation potential, they frequently express markers that overlap with those of lineage-committed or mature cell populations (Figure 1). For example, cytokeratin (CK) 19 and CK7 [30,31], commonly used to identify cholangiocytes, are also considered putative hepatic stem cell markers. Similarly, thymocyte differentiation antigen 1 (Thy-1) and receptor tyrosine kinase Kit [32,33], well-established hematopoietic stem cell markers, have been implicated in LPC identification. LPCs also share several markers with cancer stem cells. Among these, leucine-rich repeat-containing G protein-coupled receptor 5 (LGR5) and cluster of differentiation 44 (CD44) have been identified as reliable LPC markers [34,35]. Their expression is also associated with the stemness and tumorigenic potential of cancer stem cells. While the lack of specific and exclusive LPC markers continues to pose challenges and controversy in their identification, a subset of markers, including SRY-box transcription factor 9 (Sox9), muscle pyruvate kinase (MPK), oval cell marker (OV6) in human and rats, and A6 in mice, is widely recognized as representative of hepatic stem cell populations [36,37,38,39]. Hence, establishing standardized cross-model marker panels, combined with techniques such as cell sorting and genetic lineage tracing, is essential for a more accurate characterization of LPCs.

LPCs are scarce in the healthy liver but expand in response to severe and/or chronic liver injury [22,29]. Despite their significant expansion in the injured liver, the precise origin of LPCs remains unclear. Studies report that LPCs express Thy-1 and the receptor tyrosine kinase Kit—markers of hematopoietic stem cells—at the mRNA and protein levels, thereby revealing bone marrow as a potential source [40,41]. However, more recent evidence suggests that LPCs also arise from intrahepatic sources [42]. Given the complex cellular composition of the liver, various hepatic cell types—including hepatocytes, cholangiocytes, and HSCs—exhibit phenotypic and functional heterogeneity and have been identified as potential sources of LPCs [13,14,15,16,17,18]. A growing body of research supports the view that LPCs arise from hepatic cells. Therefore, this review focuses on the intrahepatic origin of LPCs.

## 3. Origin of Liver Progenitor Cells in Liver Regeneration

### 3.1. Dedifferentiation of Hepatocytes into Liver Progenitor Cells

Hepatocytes are highly specialized with minimal plasticity, primarily contributing to tissue repair through self-replication under normal physiological conditions [43,44]. However, emerging evidence indicates that hepatocytes can adopt a progenitor-like state under specific conditions, including in vitro culture, PHx, and chronic liver injury [13,14,45]. Chen et al. [13]. report that mature hepatocytes isolated from rat liver dedifferentiate into LPCs in vitro. Although significant cell death occurs during hepatocyte culture, a subset survives, progressively losing the expression of hepatocyte markers, including albumin and hepatocyte nuclear factor 1 α (HNF1α), while acquiring the expression of LPC markers, such as CK7 and OV6, in a time-dependent manner. By day 12, hepatocyte markers were undetectable, and only LPC markers were expressed. Upon transplantation into PHx liver, these cells successfully engraft and redifferentiate into hepatocytes. Hepatocyte-to-LPC dedifferentiation is also observed in in vivo models subjected to PHx or chronic liver injury. In rats undergoing PHx following retrorsine treatment, which blocks hepatocyte proliferation, hepatocytes show LPC-like morphology—including small basophilic nuclei and highly vacuolated cytoplasm—and these cells are significantly more proliferative than that of other hepatocytes [45]. Lineage-tracing studies show that hepatocytes dedifferentiate into LPCs in cholestatic liver induced by a 3,5-diethoxycarbonyl-1,4-dihydrocollidine (DDC) diet. In DDC-fed mice, hepatocytes express stem cell markers such as Sox9, A6, and secreted phosphoprotein 1 during liver injury. These hepatocyte-derived LPCs subsequently redifferentiate into mature hepatocytes during the recovery phase after DDC withdrawal. Collectively, these findings suggest that hepatocytes can undergo metaplastic reprogramming into LPCs, thereby transiently contributing to the LPC pool to support liver regeneration [14].

Hepatocyte dedifferentiation into LPCs is orchestrated by a complex network of signaling pathways that mediate cellular plasticity and fate decisions (Figure 2). Among them, the Hippo signaling pathway plays a key role in hepatocyte-to-LPC conversion. This pathway regulates cell proliferation and organ size across various tissues [46,47]. Large tumor suppressor kinases (LATS)1/2 phosphorylate yes-associated protein (YAP) and transcriptional coactivator with PDZ-binding motif (TAZ), thereby inactivating YAP/TAZ and activating Hippo signaling [48,49]. Yimlamai et al. [50]. show that YAP overexpression in hepatocytes causes loss of mature hepatocyte identity and progressive acquisition of progenitor-like gene expression profiles, as revealed by microarray analysis. When YAP overexpression ceases in these cells, hepatocyte-to-LPC dedifferentiation is reversed and hepatocyte-specific gene expression is restored. Deletion of LATS1/2 overactivates YAP/TAZ in hepatocytes, resulting in LPC expansion [51]. B-cell lymphoma 3 (Bcl3), a transcriptional coregulator of nuclear factor kappa-light-chain-enhancer of activated B cells, is also suggested as a regulator of YAP in PHx liver. In this study, upregulated Bcl3 interacts with YAP, inhibits its ubiquitination, and stabilizes YAP in hepatocytes. This Bcl3-mediated stabilization of YAP promotes the conversion of mature hepatocytes into Sox9-expressing LPCs [52].

Hedgehog (Hh), a key embryonic developmental pathway [53,54], is also activated in injured hepatocytes [55]. It is triggered when Hh ligands, including Sonic Hh (Shh), Indian Hh (Ihh), and desert Hh, bind to the canonical receptor Patched (PTC). The binding relieves PTC inhibition of smoothened (SMO), thereby allowing SMO activation. Activated SMO subsequently promotes nuclear translocation of glioma-associated oncogene homolog (GLI) transcription factors—particularly GLI1/GLI2-—which induce expression of Hh-responsive genes such as *GLI1*, *PTC*, *cyclin D*, and *cyclin E* [56]. Hh signaling promotes the growth and viability of human and rodent LPCs. In an early study using adult Ptc-lacZ reporter mice, β-galactosidase-positive Ptc-expressing cells were observed in the periportal zone [57]. These cells are smaller than adjacent mature hepatocytes and express the hepatic progenitor marker pan-CK, suggesting their identity as LPCs. In human liver tissue from patients with primary biliary cholangitis, OV6-positive LPCs coexpress GLI2 and PTC, indicating that LPCs are Hh-responsive [58]. In leptin-deficient ob/ob mice treated with ethionine, activated Hh signaling increases expression of LPC markers, including MPK and A6 [59]. The Hh pathway is also activated after PHx and is accompanied by an expansion of LPCs. Gene expression profiling after PHx reveals dynamic regulation of Hh pathway components—such as *Ihh*, *Shh*, *Smo*, *Gli1*, and *Gli2*—consistent with pathway activation. Along with Hh activation, expression of hepatic progenitor markers, such as alpha fetoprotein (AFP), fibroblast growth factor-inducible 14 (Fn14), AE1/AE3, and MPK, progressively increases between 12 and 48 h post PHx. Furthermore, primary hepatocytes isolated from the liver 48 h after PHx express Gli2 and AFP, indicating an LPC-like phenotype. Cyclopamine, a pharmacological Hh inhibitor, significantly reduces the hepatic progenitor marker levels in PHx mice [60]. Collectively, these findings highlight the critical role of Hh signaling in the emergence and maintenance of LPCs during liver injury and regeneration.

Hh signaling extends beyond hepatocytes; it also functions in HSCs, where HSC-derived paracrine signals influence hepatocyte transition into LPCs [61]. Conditional deletion of SMO in HSCs eliminates their Hh responsiveness and consequently reduces the production of Hedgehog ligands [18]. Hepatocytes isolated from mice with HSC-specific *Smo* deletion exhibit reduced proliferation, fewer progenitor-like characteristics, and higher expression of mature hepatocyte markers than those of hepatocytes from control mice with intact Hh signaling in HSCs. Loss of the signaling axis drives hepatocytes toward terminal differentiation, suggesting that HSC-derived Hh signaling is essential for maintaining hepatocyte proliferation and progenitor-like phenotype [61]. Overall, these findings highlight the critical role of Hh signaling in hepatocyte reprogramming: acting directly in hepatocytes and indirectly through HSC-mediated paracrine mechanisms.

The wingless-type MMTV integration site family (Wnt)/β-catenin signaling pathway is a major morphogen in embryogenesis and regulates hepatobiliary development, maturation, and zonation [62,63,64]. Wnt signaling begins when extracellular ligands (such as Wnt1, Wnt3a, and Wnt7a) bind to membrane-bound receptors, the frizzled family and the coreceptors low-density lipoprotein-related receptors 5 and 6. This binding stabilizes β-catenin, which translocates to the nucleus and activates target genes, including *matrix metalloproteinase* and *c-Myc* [62]. During liver regeneration after PHx in rats, β-catenin accumulates in the cytoplasm and nucleus, leading to rapid activation of the Wnt/β-catenin signaling pathway [65]. In DDC-treated mice overexpressing β-catenin, the number of A6-positive hepatocytes significantly increase [66]. Moreover, mice fed a DDC diet and genetically modified to express a nondegradable form of β-catenin show more Sox9-positive cells than those of wild-type controls [67]. Wnt7A also induces *Sox9* expression in hepatocytes and markedly enhances β-catenin-dependent transcription, indicating that canonical Wnt signaling facilitates hepatocyte reprogramming. Inhibition of Wnt ligand secretion via liver-specific knockout (KO) of *wntless*—a protein critical for Wnt ligand release—significantly reduces A6- and CK19-positive cells following DDC diet and increases mortality. Therefore, these findings suggest that Wnt/β-catenin signaling is essential for converting hepatocytes into LPCs and enabling effective liver regeneration.

Beyond the major signaling pathways mediating hepatocyte-to-LPC dedifferentiation, other signaling factors also contribute. Interleukin-6 (IL-6), a pleiotropic cytokine [68], has recently been shown to induce the conversion of primary hepatocytes into LPC-like cells [69]. These cells exhibit elevated Sox9 expression and reduced levels of mature hepatocyte markers, including albumin, hepatocyte nuclear factor 4α (Hnf4α), cytochrome P450 family 1 subfamily A member 2, and cytochrome P450 family 2 subfamily C member 9. In DDC-induced liver injury models, elevated IL-6 activates the Janus kinase/signal transducer and activator of transcription 3 (STAT3) pathway, upregulating *Sox9*, which plays a key role in maintaining the progenitor-like state [70]. These findings underscore the involvement of IL-6–mediated signaling in hepatocyte reprogramming during liver injury.

Tumor necrosis factor-like weak inducer of apoptosis (TWEAK), a macrophage-derived cytokine and its receptor, Fn14, form a signaling axis known to promote liver inflammation and fibrosis [71,72]. Recently, the TWEAK/Fn14 pathway has been implicated in the activation and proliferation of LPCs, with evidence showing it may drive hepatocyte-to-LPC conversion during liver injury. Following PHx, Fn14-positive hepatocytes accumulate in regenerating liver tissue, and hepatic *TWEAK* expression increases markedly [73]. Disrupting TWEAK/Fn14 signaling—via *Fn14* KO or administration of anti-TWEAK antibodies—reduces AFP-positive and LGR5-positive LPCs and severely impairs liver regeneration. These findings indicate that TWEAK signaling directly expands Fn14-expressing LPCs during regeneration after PHx. Consistent with these findings, *Fn14* expression in human patients with alcohol-associated hepatitis is approximately 10-fold higher than in normal controls and mainly localized to a subset of hepatocytes and LPCs [74]. Although direct lineage tracing is not possible in human samples, the concurrent LPC expansion and Fn14 upregulation strongly support TWEAK/Fn14 involvement in hepatocyte-to-LPC reprogramming in chronic liver disease.

Advances in liver regeneration research have uncovered diverse mechanisms underlying hepatocyte dedifferentiation into LPCs, and suggest that hepatocytes are a major LPC source. Key signaling pathways—Hippo, Hedgehog, Wnt/β-catenin, IL-6/STAT3, and TWEAK/Fn14—are central regulators of this process. However, mechanisms driving hepatocyte dedifferentiation vary with liver injury type and the surrounding microenvironment and remain poorly understood. Therefore, further studies are needed to define the context-specific cues and molecular networks driving hepatocyte-to-LPC conversion, and long-term, systematic investigations are essential to fully characterize these processes.

### 3.2. Conversion of Cholangiocytes into Liver Progenitor Cells

Cholangiocytes are specialized epithelial cells of the biliary system responsible for bile modification and transport [75]. Beyond bile transport, cholangiocytes are considered a key subset of LPCs in the injured liver. Cholangiocytes proliferate through ductular reaction (DR) and serve as a source of LPCs to compensate for impaired hepatocyte proliferation [76,77]. Suppression of biliary proliferation by the toxin 4,4′-diaminodiphenylmethane inhibits LPC expansion and AFP expression in the 2-acetamidofluorene (2-AAF)/PHx model [78]. With advances in lineage-tracing technologies, systems labeling cholangiocyte-specific markers, such as hepatocyte nuclear factor 1β and CK19, have demonstrated that cholangiocytes express LPC markers and differentiate into functional hepatocytes in DDC-treated or PHx-performed livers [15,16]. Tarlow et al. [79]. also report that Sox9 labeled-cholangiocytes contribute to the LPC pool in various liver injury models, including DDC diet, choline-deficient, ethionine-supplemented (CDE) diet, and CCl_4_-induced damage. These results indicate that LPCs originate from cholangiocytes.

In a study using a multi-lineage genetic tracing approach, *Fumarylacetoacetase* (*Fah*)-deficient mice—which undergo hepatocyte senescence during liver regeneration leading to LPC emergence—show that a subset of cholangiocytes co-expresses cholangiocyte markers and A6, a known progenitor cell marker [80]. In this model, LPCs show significantly lower expression of Notch downstream target genes—including *Inhibitor of DNA binding 2*, *Inhibitor of DNA binding 1*, *Clusterin*, and *One cut homeobox 1*—compared to mature cholangiocytes, suggesting that reduced Notch activity may promote LPC activation. The role of Notch signaling in this process was further validated by genetic ablation of recombination signal binding protein for immunoglobulin kappa J region (RBPJ), a key transcriptional mediator of Notch signaling. *RBPJ* KO in *Fah*-deficient mice and pharmacological inhibition of Notch signaling using dibenzazepine, a γ-secretase inhibitor, both significantly increased the LPC population. Conversely, overexpression of a constitutively active Notch intracellular domain (NICD) in cholangiocytes of *Fah*-deficient mice significantly decreased LPC numbers. Overall, these findings indicate that inhibiting Notch signaling promotes cholangiocyte-to-LPC conversion during hepatocyte-senescent liver.

The conversion of cholangiocyte-derived LPCs into functional hepatocytes has been studied in a zebrafish model [81,82,83,84]. In an acute liver injury model with extensive hepatocyte loss, cholangiocytes express Sox9b, hematopoietically expressed homeobox, and forkhead box A3, and dedifferentiate into proliferative LPCs [83]. These cholangiocyte-derived LPCs subsequently lose their original cholangiocyte markers and gain mature hepatocyte markers, indicating transdifferentiation from cholangiocytes to LPCs and hepatocytes. Overall, zebrafish models provide strong evidence that bipotent LPCs generate hepatocytes during regeneration [83,84]. Despite evidence from various studies, the molecular mechanisms driving cholangiocyte-to-LPC conversion remain poorly understood. Hence, further studies are needed to elucidate the regulatory networks controlling this plasticity and to enhance our understanding of liver cell fate dynamics.

### 3.3. Transdifferentiation of Hepatic Stellate Cells into Liver Progenitor Cells

HSCs remain quiescent under normal physiological conditions, primarily storing vitamin A [85,86]. Following liver injury, HSCs become activated and transdifferentiate into myofibroblast-like cells that drive fibrogenesis [87,88,89]. Recent evidence shows that activated HSCs can acquire LPC-like phenotypes [17,18,90]. Primary HSCs isolated from the healthy rats, cultured in hepatocyte differentiation-inducing media, begin to express hepatic markers such as albumin and HNF4α [17]. This indicates that HSCs retain latent multipotency and lineage reprogramming potential. Hedgehog signaling is upregulated upon HSC activation and has been implicated in maintaining progenitor-like or epithelial–mesenchymal transition-like states [18]. Following PHx, activated HSCs exhibit increased Hedgehog activity, inducing the expression of progenitor markers (Lgr5, Sox9, and AFP). The transforming growth factor-β (TGF-β) pathway, known for promoting HSC activation and fibrogenesis, also drives fate conversion through mesenchymal–epithelial transition [91]. In a mouse PHx model, Chen et al. [90]. report that treatment with the TGF-β inhibitor SB-431542 suppresses Sox9 and AFP expression in HSCs, indicating the role of TGF-β signaling in progenitor-like transitions.

Although studies show that HSCs acquire progenitor-like features under certain conditions, it remains unclear whether this dedifferentiation contributes functionally to liver regeneration in vivo. Genetic lineage-tracing studies provide limited evidence, and the regenerative potential of HSC-derived LPC-like cells remains uncertain, warranting further investigation.

## 4. Liver Progenitor Cells Differentiation into Mature Hepatic Cells During the Repair Process

During liver repair, LPCs differentiate into hepatocytes, cholangiocytes, or HSCs, as dictated by the liver microenvironment [10,19]. Differentiation is tightly regulated by multiple signaling pathways, including Notch, Wnt/β-catenin, and hepatocyte growth factor (HGF) (Figure 3). Notch signaling inhibits LPC-to-hepatocyte differentiation while promoting biliary fate. Ligand-activated Notch receptors undergo sequential proteolysis, releasing the NICD. NICD translocates into the nucleus and interacts with RBPJ, one of its nuclear effectors [92,93]. Liver-specific KO of *RBPJ* caused abnormal biliary development and impaired regeneration in PHx livers [94]. Tang et al. [95]. investigated the role of epithelial cell adhesion molecule (EpCAM) as an upstream modulator of Notch signaling in human hepatocyte-derived LPCs. To induce LPC-like characteristics from hepatocytes, human primary hepatocytes were cultured in a transition and expansion medium containing epidermal growth factor (EGF), HGF, Wnt and YAP activators, TGF-β inhibitor, and Rho-associated coiled-coil kinase inhibitors. Under these conditions, hepatocyte markers (albumin and HNF4A) gradually decreased, while LPC markers (CK19, CK7, and Sox9) increased, indicating successful hepatocyte-to-LPC conversion. Hepatocyte-derived LPCs were transduced with *EpCAM*-shRNA lentivirus to silence *EpCAM* expression. This reduced Notch1 expression and increased hepatocyte markers (Cytochrome P450 family 3 subfamily A member 4 and albumin), indicating that EpCAM promotes Notch signaling and inhibits LPC-to-hepatocyte differentiation. Notch1, induced by epidermal growth factor receptor (EGFR) signaling, also promotes LPC-to-cholangiocyte differentiation, as shown using an LPC line established by Kitade et al. [96]. Among LPCs isolated from *Egfr*^flox/flox^ mice fed a DDC diet, EpCAM-positive cells lacking hematopoietic markers were cultured under non-adherent conditions to enrich self-renewing cells, generating the *Egfr*^flox/flox^ LPC line. The LPC line transduced with Ad-CMV-Cre to delete *EGFR* exhibited reduced Notch1 levels. Restoring EGFR in the LPC line recovers Notch1 expression and induces cholangiocyte-like branching. LPCs isolated from DDC-fed mice had high Notch1 and Notch2 levels, and the adjacent myofibroblasts expressed Jagged-1, a Notch ligand [97]. Intravenous administration of a γ-secretase inhibitor, which blocks Notch receptor cleavage and activation, to DDC-fed mice reduced LPC numbers and decreased expression of Notch downstream targets and biliary-associated genes. These findings suggest that Notch signaling suppresses LPC-to-hepatocyte differentiation while promoting LPC-to-cholangiocyte differentiation.

Wnt/β-catenin signaling mediates LPC differentiation into hepatocytes. *Wnt1* deletion blocked β-catenin nuclear translocation and prevented LPC differentiation into hepatocytes in 2-AAF/PHx rats [98]. Activating the canonical Wnt pathway by inhibiting β-catenin degradation in DDC-fed mice enhanced β-catenin nuclear accumulation in LPCs and promoted their hepatocyte differentiation [97]. This study showed macrophage-derived Wnt3a as a key inducer of Wnt signaling in LPCs. Mouse bone marrow–derived macrophages upregulated canonical Wnt ligands Wnt3a and Wnt7a upon phagocytic activation. Co-culturing LPCs with macrophages treated with the Wnt3a inhibitor reduced hepatocyte marker *Hnf4α* and *Hnf1α* expression in LPCs. These results were further confirmed in CDE-fed mice. Macrophage ablation in CDE diet-fed mice decreased β-catenin nuclear accumulation and *Hnf4α* and *Hnf1α* expression in LPCs, impairing their hepatocyte transition.

Wnt signaling appears to contribute to both the reprogramming of hepatocytes into LPCs and the differentiation of LPCs into hepatocytes. Although different Wnt ligands have been implicated in these seemingly opposite processes—such as Wnt7a in hepatocyte dedifferentiation into LPCs and Wnt1/Wnt3a in LPC differentiation toward hepatocytes—they all activate the β-catenin signaling pathway [97]. However, experimental evidence explaining how Wnt signaling can mediate these opposing outcomes remains limited. This discrepancy may be attributed to differences in the type or severity of liver injury, as well as the timing of observation during liver regeneration. Therefore, further in-depth studies are required to elucidate the precise role of Wnt signaling in the dynamic transition between hepatocytes and LPCs, taking into account the diverse contextual factors that may influence these outcomes.

The HGF signaling pathway is well-known to promote LPC commitment toward the hepatocytic lineage [99]. HGF binding to its receptor, tyrosine kinase mesenchymal–epithelial transition factor (c-MET), activates multiple intracellular signaling cascades. c-MET directly activates the phosphoinositide 3-kinase (PI3K)/protein kinase B (AKT) and STAT3 pathways [100,101]. In 2-AAF/PHx-rats, exogenous HGF treatment induced an early upregulation of AFP by day 4 after PHx and increased the number of AFP-positive LPCs in the liver [102]. *Albumin* transcripts markedly increased between days 8 and 12 post PHx in these rats. c-MET also plays a critical role in regulating LPC differentiation. *c-Met* deletion in LPCs blocked HGF-induced STAT3 phosphorylation and prevented hepatocyte-like morphology or expression of hepatocyte markers even after HGF treatment [99]. However, restoring c-MET expression via gene reintroducing recovered STAT3 phosphorylation and upregulated hepatocyte differentiation genes, including *albumin* and *HNF4α* [96]. These findings suggest that HGF signaling serves as a crucial regulator accelerating the transition of LPCs into hepatocytes.

TGF-β, non-canonical Wnt, and Hh signaling have been reported to induce LPC differentiation into HSCs [103,104,105]. Studies show that LPC-derived HSCs secrete HGF and produce ECM to maintain tissue architecture [12,106]. However, many consider LPC-derived HSCs a contributor to liver fibrosis rather than a facilitator of regeneration [103,104,105]. Since this review focuses on the role of LPCs in liver regeneration, the signaling mechanisms of LPC-to-HSC differentiation were not discussed. Nonetheless, further studies are needed to elucidate the role of LPC-derived HSCs in liver regeneration and to determine whether this differentiation is regenerative or fibrogenic.

## 5. Conclusions

LPCs are essential for liver regeneration, particularly when the regenerative capacity of hepatocytes is compromised [8,9]. These bipotent cells are predominantly activated in response to liver injury and can differentiate into hepatocytes and cholangiocytes during repair processes [10,19]. Lineage tracing and transcriptomic analyses revealed that LPCs can originate from mature hepatocytes via dedifferentiation, cholangiocytes via transdifferentiation, and possibly HSCs under specific pathological conditions [13,14,15,16,17,18]. Table 1 summarizes the reported sources of LPCs according to the type of liver injury, timing, associated signaling pathways, and supporting experimental evidence. These processes highlight liver cellular plasticity and underscore the dynamic nature of LPC origin.

Despite extensive research, the precise definition of LPC populations remains challenging due to the lack of unique, definitive markers and their phenotypic overlap with mature hepatic cells. Moreover, the extent to which LPCs contribute to liver regeneration in human pathological conditions is still not clearly understood. Most mechanistic insights into LPC biology have been derived from rodent models, where genetic lineage-tracing enables precise tracking of LPC fate [107]. However, such approaches cannot be applied in human studies, as genetic manipulation and long-term lineage tracing are neither ethically permissible nor technically feasible in clinical settings [107,108]. Furthermore, humans exhibit substantial genetic and physiological variability influenced by several factors, such as sex, age, underlying health conditions, and environmental exposures. This inherent heterogeneity significantly complicates the direct extrapolation of findings from animal models to human liver regeneration and has hindered the delineation of well-defined regenerative mechanisms in humans. Another critical limitation lies in the inconsistent and sometimes contradictory findings regarding the involvement of specific signaling pathways in LPC-mediated regeneration. The scarcity of in-depth studies focused on dissecting the molecular mechanisms governing LPC activation, transition, and differentiation further obscures our understanding of their precise functions.

To address these limitations, more comprehensive approaches are required. The identification and validation of robust and specific LPC markers are essential for advancing our understanding of their cellular dynamics and lineage potential in human liver. Thus, systemic reviews and meta-analyses of both animal and human studies could be helpful to overcome current limitations. Recent advances in big data-driven bioinformatics offer powerful tools for resolving current gaps [109,110,111]. The integration of large-scale, high-dimensional datasets—such as single-cell RNA sequencing, spatial transcriptomics, and multi-omics profiling—provides unprecedented resolution into LPC heterogeneity, lineage relationships, and functional states during liver regeneration. In addition, cross-species comparative analyses enabled by these technologies may help bridge the translational gap between murine and human models. Such approaches can facilitate the identification of conserved versus species-specific pathways, highlight the contexts in which animal data remain clinically relevant, and guide the refinement of preclinical experimental models. Based on these technological and analytical advancements, future studies should focus on clarifying the context-specific roles of LPCs in various types of liver injury and the long-term fate and functionality of LPC-derived cells. Comprehensive characterization of LPC dynamics may enable new therapeutic approaches to enhance liver regeneration and treat chronic liver disorders, including fibrosis, cirrhosis, and liver failure.

## Figures and Tables

**Figure 1 biology-14-01361-f001:**
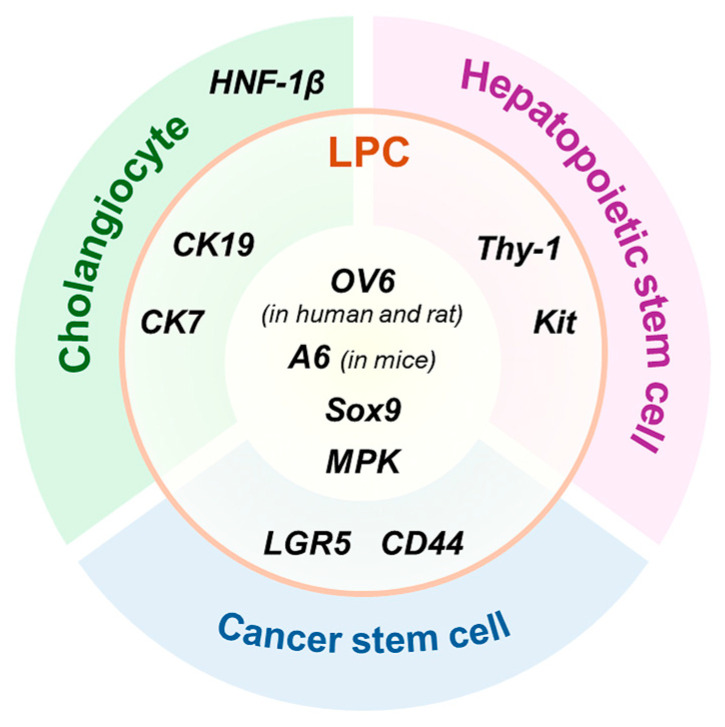
Shared and distinct markers of liver progenitor cells (LPCs) and related cell types. LPCs are heterogeneous, and their markers often overlap with those of various other cell types, including cholangiocytes, hematopoietic stem cells, and cancer stem cells. OV6 in rats, A6 in mice, SRY-box transcription factor 9 (Sox9), and muscle pyruvate kinase (MPK) are widely recognized as representative markers of LPCs. Cytokeratin (CK) 19 and CK7, originally identified as cholangiocyte markers, have also been extensively used to identify LPCs, whereas hepatocyte nuclear factor-1β (HNF-1β) is considered a cholangiocyte-specific marker. In addition, hematopoietic stem cell markers such as thymocyte differentiation antigen 1 (Thy-1) and receptor tyrosine kinase Kit (Kit), as well as cancer stem cell markers such as leucine-rich repeat-containing G protein-coupled receptor 5 (LGR5) and cluster of differentiation 44 (CD44), are also regarded as reliable markers of LPCs.

**Figure 2 biology-14-01361-f002:**
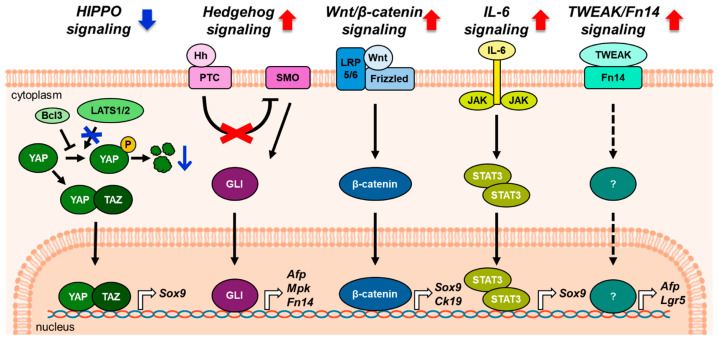
Signaling pathways driving hepatocyte dedifferentiation into LPCs in injured liver. Dedifferentiation of hepatocytes into LPCs occurs in response to severe and/or chronic liver injury, such as 2-acetamidofluorene (2-AAF)-treated partial hepatectomy, a choline-deficient, ethionine-supplemented (CDE) diet, or a 3,5-diethoxycarbonyl-1,4-dihydrocollidine (DDC) diet. This process is regulated by several key signaling pathways, including Hippo, Hedgehog (Hh), Wingless-type MMTV integration site family (Wnt)/β-catenin, interleukin-6 (IL-6)/signal transducer and activator of transcription 3 (STAT3), and tumor necrosis factor-like weak inducer of apoptosis (TWEAK)/fibroblast growth factor-inducible 14 (Fn14) signaling. Inactivation of the Hippo pathway—via inhibition of large tumor suppressor kinases (LATS) and activation of B-cell lymphoma 3 (Bcl3)—leads to stabilization of yes-associated protein (YAP), which interacts with the transcriptional coactivator with PDZ-binding motif (TAZ). The YAP/TAZ complex translocates into the nucleus and induces expression of *Sox9*, promoting hepatocyte dedifferentiation. Hh signaling is initiated by the binding of Hh ligands to Patched (PTC), relieving inhibition of Smoothened (SMO) and enabling GLI transcription factors to translocate into the nucleus. GLI activation upregulates expression of *alpha-fetoprotein* (*Afp*), *Mpk*, and *fibroblast growth factor-inducible 14* (*Fn14*). Wnt/β-catenin signaling is triggered by the binding of Wnt ligands to Frizzled and low-density lipoprotein-related receptors 5 and 6 (LRP5/6) receptors, resulting in β-catenin stabilization and nuclear translocation. β-catenin then activates downstream targets such as *Sox9* and *CK19*. In addition, IL-6 signaling activates the Janus kinase (JAK)/STAT3 pathway, further enhancing Sox9 expression. TWEAK, a cytokine that binds Fn14, promotes expression of *Afp* and *Lgr5*, although the exact mechanisms by which TWEAK contributes to hepatocyte dedifferentiation remain unclear. Dashed lines indicate pathways that are not yet fully elucidated.

**Figure 3 biology-14-01361-f003:**
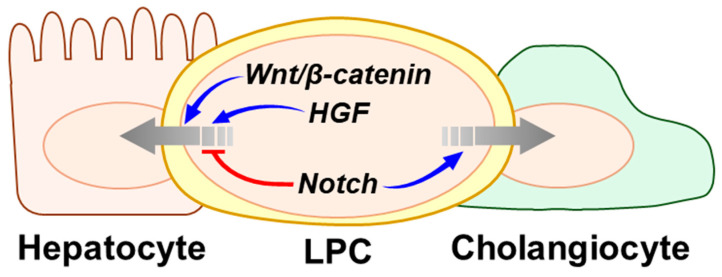
Signaling pathways regulating LPC differentiation during liver regeneration. LPC fate is determined by multiple signaling pathways, including Notch, Wnt/β-catenin, and Hepatocyte growth factor (HGF). Upregulation of Wnt/β-catenin and HGF signaling in LPCs promotes differentiation toward the hepatocytic lineage. In contrast, activation of Notch signaling in LPCs preferentially induces cholangiocytic differentiation, whereas its inhibition facilitates hepatocytic lineage specification.

**Table 1 biology-14-01361-t001:** Context-dependent sources of LPCs: injury, timing, signaling pathway, and evidence.

Source	Species	Injury	Timing	Signaling	Validation Method
Hepatocyte	Human	Alcoholic hepatitis	ABIC score ≥ 6.71 points	TWEAK/Fn14	Liver biopsy
	Mouse	70% PHx	24, 48 h after PHx	TWEAK/Fn14	Systemic *Fn14* deletion
			3 h after PHx	Hippo	Systemic *Bcl3* deletion
			12, 24, 48 h after PHx	Hedgehog	Immunohistochemistry and Western blot
			12, 24, 48 h after PHx	Hedgehog	Isolated pHeps from HSC-specific *Smo* deleted mice
		0.1% DDC	150 days of DDC diet	Wnt/β-catenin	Hepatocyte-specific *β-catenin* overexpression
			2 wks or 1 month of DDC diet	Wnt/β-catenin	Liver-specific *Wls* or *Lrp5/6* deletion
			1 wks of DDC diet	IL-6	Hepatocyte-specific *IL-6* deletion
Cholangiocyte	Mouse	70% PHx	7 days after PHx	-	HNF1β lineage tracing
		MCD and 50% PHx	7–14 days of MCD diet; 48 h after PHx	-	CK19 lineage tracing and hepatocyte-specific p21 overexpression
		0.1% DDC	14 days for recovery after 12 days of DDC diet	-	CK19 lineage tracing and hepatocyte-specific *β1 integrin* deletion
			3 wks of DDC diet	Notch	CK19 lineage tracing and systemic *Fah* deletion
			4 wks of DDC diet	-	SOX9 lineage tracing
		CDE	2 wks for recovery after 3 wks of CDE diet	-	SOX9 lineage tracing
		CCl_4_	1 wks for recovery after 5 wks of CCl_4_	-	SOX9 lineage tracing
	Zebrafish	Hepatocyte ablation	8 and 24 h after hepatocyte ablation	-	NTR-Mtz system
			24 h after hepatocyte ablation	-	NTR-Mtz system
HSC	Mouse	70% PHx	24, 48, 72 h after PHx	Hedgehog	SMO lineage tracing
			1, 3 days after PHx	TGF-β	Immunohistochemistry

ABIC score, Age, serum bilirubin, international normalized ratio, and serum creatinine score; TWEAK, tumor necrosis factor-like weak inducer of apoptosis; Fn14, fibroblast growth factor-inducible 14; h, hours; PHx, partial hepatectomy; Bcl3, B-cell lymphoma 3; pHeps, primary hepatocytes; HSC, hepatic stellate cell; SMO, Smoothened; DDC, 3,5-diethoxycarbonyl-1,4-dihydrocollidine; wks, weeks; Wnt, wingless-type MMTV integration site family; Wls, Wntless; Lrp5/6, low-density lipoprotein-related receptors 5 and 6; IL-6, interleukin-6; HNF1β, hepatocyte nuclear factor 1 β; MCD diet, methionine- and choline-deficient diet; CK19, cytokeratin 19; Fah, Fumarylacetoacetase; SOX9, SRY-box transcription factor 9; CDE diet, choline-deficient, ethionine-supplemented diet; CCl_4_, carbon tetrachloride; NTR-Mtz system, nitroreductase–metronidazole system; TGF-β, transforming growth factor-β.

## Data Availability

Not applicable.

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
