# Peer review of "Liver Progenitor Cells: Cellular Origins, Plasticity, and Signaling Pathways in Liver Regeneration"

_biology, 2025, doi:10.3390/biology14101361_

Round 1

Reviewer 1 Report

Comments and Suggestions for Authors

The authors, Han et al., have submitted a comprehensive, well-written manuscript entitled "Liver Progenitor Cells: Cellular Origins, Plasticity, and Signal-2 Pathways in Liver Regeneration." The authors did a great job organizing the review and writing it very well. A couple of inconsistencies may be addressed before accepting the review:

The authors need to provide an expansion of the protein acronyms OV6 and A6 on line 101.

The expanded form of mouse and human protein names should not be italicized. The authors need to revise several instances of this throughout the manuscript.

Author Response

The authors, Han et al., have submitted a comprehensive, well-written manuscript entitled "Liver Progenitor Cells: Cellular Origins, Plasticity, and Signal-2 Pathways in Liver Regeneration." The authors did a great job organizing the review and writing it very well. A couple of inconsistencies may be addressed before accepting the review:

1. The authors need to provide an expansion of the protein acronyms OV6 and A6 on line 101.

: We appreciate your comments. In the revised manuscript, we have provided the full name of OV6 on line 101 as requested. A6 is not an abbreviation. It represents an uncharacterized epitope associated with mouse hepatic oval cells. Therefore, we were unable to provide an expanded form for A6.

2. The expanded form of mouse and human protein names should not be italicized. The authors need to revise several instances of this throughout the manuscript.

: We carefully reviewed the entire manuscript to ensure that gene names are consistently italicized, whereas protein names were not. The revised portions are highlighted in the revised version.

Reviewer 2 Report

Comments and Suggestions for Authors

My comments are as follows:

  1. The manuscript is generally well written and informative.
  2. The iThenticate similarity index is currently 27%. Please revise the manuscript to reduce the similarity below 20%.
  3. Line 443: Please clarify what the term "6. patent" refers to in this context.
  4. Consider including a discussion on the relative contributions of hepatocytes, cholangiocytes, and hepatic stellate cells (HSCs) to the liver progenitor cell (LPC) pool during different types of liver injury.
  5. It would strengthen the manuscript to discuss epigenetic reprogramming or master transcription factors involved in lineage shifts. Please consider including additional transcriptional regulators in this context.
  6. Consider elaborating on the gaps in the current literature and suggesting possible future directions to further enhance the scientific soundness and relevance of the review.

Author Response

My comments are as follows:

1. The manuscript is generally well written and informative.

: Thank you for your comment.

2. The iThenticate similarity index is currently 27%. Please revise the manuscript to reduce the similarity below 20%.

: We appreciate the reviewer’s valuable comment regarding the similarity index. To address this concern, we carefully evaluated the iThenticate report. According to the report, the majority of the matched sources showed less than 1% similarity each and primarily originated from the use of standard terminology and full names of specific terms. When we applied the filter option excluding matches <1%, which is widely accepted in academic publishing to avoid inflation from common phrases and fixed terminology that are used frequently in this field, the resulting similarity index is 4%. We respectfully ask for the reviewer’s understanding that this value satisfies the requirement of remaining well below 20%. For your reference, we have attached the iThenticate report showing that similarity index of our manuscript is 4% with filter option excluding matches <1%.

3. Line 443: Please clarify what the term "6. patent" refers to in this context.

: We apologize for any confusion caused by this section. As there are no patents associated with this study, we have revised the section to state “Not applicable.” to prevent any misunderstanding.

4. Consider including a discussion on the relative contributions of hepatocytes, cholangiocytes, and hepatic stellate cells (HSCs) to the liver progenitor cell (LPC) pool during different types of liver injury.

: Thank you for your helpful comments. In the revised manuscript, we added the table (Table1) that summarized and compared the relative contributions of hepatocytes, cholangiocytes, and hepatic stellate cells to the LPC pool under different types of liver injury. By summarizing the relevant references in a table, the contributions of each cell type to the LPC pool under various injury conditions are presented more clearly, thereby providing additional clarity and strengthening the overall discussion.

5. It would strengthen the manuscript to discuss epigenetic reprogramming or master transcription factors involved in lineage shifts. Please consider including additional transcriptional regulators in this context.

: We sincerely appreciate the reviewer’s thoughtful suggestion. In this review, we focused on the signaling pathways governing intrahepatic cell conversion into liver progenitor cells (LPCs), as well as the subsequent differentiation of LPCs into functional hepatic cells. While we fully recognize the critical importance of epigenetic reprogramming and master transcription factors in lineage conversion, these topics lie outside our core area of expertise. We are concerned that attempting to review these complex mechanisms without sufficient domain knowledge could lead to the unintended dissemination of inaccurate or oversimplified information. Furthermore, covering epigenetic reprogramming, master transcription factors, and all related aspects in the single review would extend beyond the intended scope of this manuscript. We therefore believe that these important topics are better suited for a dedicated future review by experts in the field, where they can be explored in the depth and detail they warrant.

6. Consider elaborating on the gaps in the current literature and suggesting possible future directions to further enhance the scientific soundness and relevance of the review.

: As suggested, we have elaborated on the limitations of current studies and outlined possible future directions in the conclusion section to further strengthen the scientific soundness and relevance of the review: “Despite extensive research, the precise definition of LPC populations remains challenging due to the lack of unique, definitive markers and their phenotypic overlap with mature hepatic cells. Moreover, the extent to which LPCs contribute to liver regeneration in human pathological conditions is still not clearly understood. Most mechanistic insights into LPC biology have been derived from rodent models, where genetic lineage-tracing enables precise tracking of LPC fate. However, such approaches cannot be applied in human studies, as genetic manipulation and long-term lineage tracing are neither ethically permissible nor technically feasible in clinical settings. Furthermore, humans exhibit substantial genetic and physiological variability influenced by several factors, such as sex, age, underlying health conditions, and environmental exposures. This inherent heterogeneity significantly complicates the direct extrapolation of findings from animal models to human liver regeneration and has hindered the delineation of well-defined regenerative mechanisms in humans. Another critical limitation lies in the inconsistent and sometimes contradictory findings regarding the involvement of specific signaling pathways in LPC-mediated regeneration. The scarcity of in-depth studies focused on dissecting the molecular mechanisms governing LPC activation, transition, and differentiation further obscures our understanding of their precise functions.

       To address these limitations, more comprehensive approaches are required. The identification and validation of robust and specific LPC markers are essential for advancing our understanding of their cellular dynamics and lineage potential in human liver. Thus, systemic reviews and meta-analyses of both animal and human studies could be helpful to overcome current limitations. Recent advances in big data-driven bioinformatics offer powerful tools for resolving current gaps. The integration of large-scale, high-dimensional datasets—such as single-cell RNA sequencing, spatial transcriptomics, and multi-omics profiling—provides unprecedented resolution into LPC heterogeneity, lineage relationships, and functional states during liver regeneration. In addition, cross-species comparative analyses enabled by these technologies may help bridge the translational gap between murine and human models. Such approaches can facilitate the identification of conserved versus species-specific pathways, highlight the contexts in which animal data remain clinically relevant, and guide the refinement of preclinical experimental models. Based on these technological and analytical advancements, future studies~~~”

Reviewer 3 Report

Comments and Suggestions for Authors

This review reported recent literature showing that liver progenitor cells (LPCs) are a heterogeneous, bipotent population activated when hepatocyte replication is impaired, and that LPCs can originate intrahepatically via dedifferentiation or transdifferentiation of mature hepatic cell types, involving different signaling pathways, such as YAP/TAZ, Hedgehog, Wnt/β‑β-catenin, et al.  It suggests that a more detailed characterization of LPC dynamics can inform regenerative therapies. This is good work on recent advance on LPC and will help to understand the role of LPCs in liver disease.

Some weaknesses need to be improved before this work can be published.

1 The manuscript repeatedly states that hepatocytes, cholangiocytes, or HSCs can serve as LPC sources, but does not systematically weigh their relative contributions across different injury contexts. Please add a table that maps injury model → primary reported LPC source(s) → strength/type of evidence, with notes on timing and species. This will help readers to understand context-specific predominance rather than present a uniform conclusion.

2 Please include a section explicitly contrasting animal vs human evidence, discuss limitations, and propose translational strategies.

3 Several pathways are invoked as promoting both dedifferentiation and redifferentiation. Please provide a dedicated subsection on signaling context and timing, proposing models, and explicitly identifying testable hypotheses for future experiments.

Overall, the review is timely, well-organized, and valuable to the field. I recommend a revision before publication that addresses the key conceptual gaps above.

Author Response

This review reported recent literature showing that liver progenitor cells (LPCs) are a heterogeneous, bipotent population activated when hepatocyte replication is impaired, and that LPCs can originate intrahepatically via dedifferentiation or transdifferentiation of mature hepatic cell types, involving different signaling pathways, such as YAP/TAZ, Hedgehog, Wnt/β‑β-catenin, et al.  It suggests that a more detailed characterization of LPC dynamics can inform regenerative therapies. This is good work on recent advance on LPC and will help to understand the role of LPCs in liver disease. Some weaknesses need to be improved before this work can be published.

1. The manuscript repeatedly states that hepatocytes, cholangiocytes, or HSCs can serve as LPC sources, but does not systematically weigh their relative contributions across different injury contexts. Please add a table that maps injury model → primary reported LPC source(s) → strength/type of evidence, with notes on timing and species. This will help readers to understand context-specific predominance rather than present a uniform conclusion.

: As you requested, we added Table 1 that summarized and compared the relative contributions of hepatocytes, cholangiocytes, and hepatic stellate cells (HSCs) to the LPC pool under different types of liver injury in the revised manuscript.  Table 1 systematically maps each liver injury model to its corresponding primary reported LPC sources. The table also provides information on validation methods, timing, and species involved in the revised manuscript.

2. Please include a section explicitly contrasting animal vs human evidence, discuss limitations, and propose translational strategies.

: As you suggested, we discuss how findings from rodent lineage-tracing studies provide mechanistic insights, whereas human studies are constrained by ethical and technical limitations. We also highlight the translational challenges that arise from these discrepancies and suggest strategies to overcome them: “Despite extensive research, the precise definition of LPC populations remains challenging due to the lack of unique, definitive markers and their phenotypic overlap with mature hepatic cells. Moreover, the extent to which LPCs contribute to liver regeneration in human pathological conditions is still not clearly understood. Most mechanistic insights into LPC biology have been derived from rodent models, where genetic lineage-tracing enables precise tracking of LPC fate. However, such approaches cannot be applied in human studies, as genetic manipulation and long-term lineage tracing are neither ethically permissible nor technically feasible in clinical settings. Furthermore, humans exhibit substantial genetic and physiological variability influenced by several factors, such as sex, age, underlying health conditions, and environmental exposures. This inherent heterogeneity significantly complicates the direct extrapolation of findings from animal models to human liver regeneration and has hindered the delineation of well-defined regenerative mechanisms in humans. Another critical limitation lies in the inconsistent and sometimes contradictory findings regarding the involvement of specific signaling pathways in LPC-mediated regeneration. The scarcity of in-depth studies focused on dissecting the molecular mechanisms governing LPC activation, transition, and differentiation further obscures our understanding of their precise functions.

       To address these limitations, more comprehensive approaches are required. The identification and validation of robust and specific LPC markers are essential for advancing our understanding of their cellular dynamics and lineage potential in human liver. Thus, systemic reviews and meta-analyses of both animal and human studies could be helpful to overcome current limitations. Recent advances in big data-driven bioinformatics offer powerful tools for resolving current gaps. The integration of large-scale, high-dimensional datasets—such as single-cell RNA sequencing, spatial transcriptomics, and multi-omics profiling—provides unprecedented resolution into LPC heterogeneity, lineage relationships, and functional states during liver regeneration. In addition, cross-species comparative analyses enabled by these technologies may help bridge the translational gap between murine and human models. Such approaches can facilitate the identification of conserved versus species-specific pathways, highlight the contexts in which animal data remain clinically relevant, and guide the refinement of preclinical experimental models. Based on these technological and analytical advancements, future studies~~~”

3. Several pathways are invoked as promoting both dedifferentiation and redifferentiation. Please provide a dedicated subsection on signaling context and timing, proposing models, and explicitly identifying testable hypotheses for future experiments.

: Thank you for your valuable comment. As you pointed out, multiple signaling pathways are intricately involved in both the dedifferentiation of hepatic cells into LPCs and the subsequent differentiation processes. To provide a clearer overview, we have summarized the relevant signaling contexts and timing in a table. In addition, we acknowledged the limitations arising from contradictory findings regarding these signaling pathways and further discussed these limitations and included several suggestions that may guide future research directions in the conclusion section.

Overall, the review is timely, well-organized, and valuable to the field. I recommend a revision before publication that addresses the key conceptual gaps above.

: Thank you for your thoughtful effort in reviewing the manuscript.